# Impact of Gut Microbiota on Aging and Frailty: A Narrative Review of the Literature

**DOI:** 10.3390/geriatrics9050110

**Published:** 2024-08-31

**Authors:** Selene Escudero-Bautista, Arianna Omaña-Covarrubias, Ana Teresa Nez-Castro, Lydia López-Pontigo, Maribel Pimentel-Pérez, Alonso Chávez-Mejía

**Affiliations:** 1Department of Gerontology, School of Medical Science, Autonomous University of the State of Hidalgo, San Agustín Tlaxiaca 42060, Hidalgo, Mexicolydial@uaeh.edu.mx (L.L.-P.); bertha_pimentel@uaeh.edu.mx (M.P.-P.); 2Department of Nutrition, School of Medical Science, Autonomous University of the State of Hidalgo, San Agustín Tlaxiaca 42060, Hidalgo, Mexico; teresa_nez@uaeh.edu.mx; 3Department of Medicine, School of Medical Science, Autonomous University of the State of Hidalgo, San Agustín Tlaxiaca 42060, Hidalgo, Mexico; ch278471@uaeh.edu.mx

**Keywords:** gut microbiota, dysbiosis, aging, frailty

## Abstract

Aging is a natural, complex, and individual process that focuses on the progressive decay of the body and a decrease in cell function that begins in approximately the sixth decade of life and ends with death. Current scientific evidence shows that the aging process is mostly related to genetic load and varies because of the environment. Therefore, aging can be adjusted through the intervention of factors that control homeostasis in genetic, biochemical, and immunological processes, including those involving the gut microbiota. Indeed, the diversity of the gut microbiota decreases during aging, based on the presence of modifications in the hormonal, immunological, and operational processes of the gastrointestinal tract. These modifications lead to a state of dysbiosis. However, altering bacterial communities remains complicated due to the great diversity of factors that influence their modification. Alterations caused by the aging process are known to foster dysbiosis and correspond to conditions that determine the degree of frailty in senior citizens. Consequently, the microbial structure can be used as a biomarker for geriatric care in the promotion of healthy aging.

## 1. Introduction

The microbiota refers to the community of living microorganisms within an ecological space. The microbiota present in a human being’s intestine, known as the gut microbiota (GM), is the most representative and diverse example [1]. The gut microbiota is involved in disease processes because of its active participation in the somatic, nutritional, and immunity processes, among others [2]. 

The composition of the GM changes throughout a human’s life, starting with gestation and the moment when the fetus is exposed to the amniotic contents, umbilical cord, and meconium located in the placental microbiota [3]. After gestation, the microbiota composition changes again based on the time of gestation, labor type, type of food, and other environmental factors such as the presence of siblings, pets, and geographic location [3]. After beginning complementary feeding and ending breastfeeding, significant changes can be observed in the intestinal makeup due to the diversity and increase in activity among bacteria [2].

The GM remains stable from infancy to puberty, reaching its maximum point of diversity during adulthood. This stage features the greatest diversity and complexity, primarily due to genetics and lifestyle [4]. Upon reaching senescence, the diversity of bacterial settlements in the GM is reduced due to hormonal and immunological changes, as well as changes to the operation of the digestive tract, leading to a state of dysbiosis [4].

This bacterial dysbiosis causes a bidirectional relationship between the gut microbiota and aging among the elderly. This process influences immune, inflammatory, hormonal, and even neuronal processes, which are related to the development of diseases at this stage of life [5,6]. The following review seeks to enrich our understanding of the repercussions of the GM during the aging process due to the impacts of the microbiota on the diverse, dynamic processes of diseases.

### 1.1. Aging and Frailty

Aging is a natural, complex, and individual process involving the gradual deterioration of the body and a reduction in cellular function that begins during the sixth decade of life and ends with death [7,8]. According to the World Health Organization, aging is classified into three periods: (1) Early Aging, from 60 to 74 years; (2) Advanced Aging, from 70 to 90 years; and (3) Longevity, 90 years and beyond [8].

Aging is characterized by a reduction in bodily functions, thereby producing the “nine features of aging”: genomic instability, telomere attrition, epigenetic alterations, loss of proteostasis, dysregulated nutrient sensing, mitochondrial dysfunction, cellular senescence, stem cell depletion, and altered intercellular communication. These conditions promote deterioration in self-regulation, structural regeneration, and functional changes in tissues and organs that lead to the expansion of chronic diseases and the appearance of frailty [7,8,9]. 

In this sense, frailty is a clinical condition considered to be the highest clinical state of vulnerability of the individual to endogenous and exogenous stressors. It is associated with an increased risk of disability, which is provoked by the predisposition to failure in homeostasis after a stressful event [10,11]. This state can lead to anorexia, sarcopenia, a reduction in cognitive functions, decreased mobility, and a lack of independence [11]. Notably, frailty has not been linked with chronological age. Loss of the GM is related to frailty, which itself is associated with the quality of aging based on the degree of decay in one’s quality of life [9,12]. 

Under this background, the current scientific evidence indicates that the aging process generally concludes due to genetic load and is influenced by the environment [8]. In this regard, the GM is considered as a relevant factor in the aging process, due to its implications in intestinal permeability and immune function, which lead to inflammation, since inflammation is the most common biological condition in aging and age-related diseases [13].

For this reason, the present research studies aging from the perspective of factors that intervene and control homeostasis of the genetic, biochemical, and immunological processes of the gut microbiota [12,14].

### 1.2. Gut Microbiota in Aging

The GM plays an essential role in aging since it helps regulate the nine characteristics of aging, as well as processes such as fiber catabolism, the biosynthesis of vitamins and amino acids, the detoxification of xenobiotics, and modulation of the immune system, among others [7,9].

As previously mentioned, the GM can adjust immune function responses to maintain inflammatory and anti-inflammatory balance. The GM is a component of the microbiota–gut–brain axis, which influences the strength and disease progression of metabolic, digestive, and mental diseases. Furthermore, the GM actively participates in genetic and epigenetic regulation and processes of nutrition [15].

However, the GM in senior citizens is different than that in young adults and has greater interindividual variety, which involves a dysbiosis state [9]. The physiological changes in aging produce dysbiosis, which is connected more strongly with physiological than chronological age. Medication, diet, physical activity, and comorbidities are also influencing factors [9].

Broadly speaking, the GM contains 1000 species of diverse bacteria corresponding to four predominant bacterial phyla: *Proteobacteria*, *Actinobacteria*, *Firmicutes*, and *Bacteroidetes*, with the last two constituting 90% of the bacterial community. Variations in the formation of the GM can be found at both the bacterial phylum level and at the species level depending on ethnicity, diet, lifestyle, and geography [10,16].

The GM was previously considered to be relatively stable in adulthood, unlike that during aging, since the GM manifests gradual changes not only in its composition but also in its activity, leading to a reduction in diversity [16].

As previously described, older adults possess a higher prevalence of *Bacteroidetes* compared with the abundance of *Firmicutes* observed in younger adults. At the species level, reduced concentrations of anaerobic bacteria have been observed, including *Clostridium* cluster XIVa, *Faecalibacterium prausnitzii*, *Akkermansia muciniphila*, *Ruminococcus bromii*, and *Ruminococcus gnavus* [16]. A decrease in *Bifidobacterium* was also reported, together with an increase in *Clostridium*, *Lactobacillus*, *Enterobacteriaceae*, and *Enterococcus* and changes in the abundance of *Bacteroides* and *Clostridium* [8].

The decreased diversity of alpha taxa such as *Eubacterium dolium*, *Erysipelotrinchaceae*, and *Prevotella* spp. and probiotic species such as *Faecalibacterium prausnitzii* has also been correlated with the degree of frailty, mental health, sarcopenia, systemic inflammation, and other related conditions [15,16,17]. Likewise, the decreased production of butyrate species such as *Faecalibacterium prausitzzi*, *Eubacterium halii*, and *Eubacterium rectalis* was inversely associated with the anabolic processes related to frailty [10].

Conversely, the increased abundances of *Bacteroides* and *Roseburia*, including *Christensenella*, *Bifidobacterium*, and *Akkermansia*, seems to affect longevity, as such taxa are common in centenarian and supercentenarian populations. Other studies have demonstrated an expansion of bacteria such as *Alistipes putredinis* and *Odoribacter splanchnicus* in the same population [8,16].

Due to the above, the development and application of analytical tools through complete genomic sequencing are currently being studied. With the aim of using the GM as a predictor model for aging, and for the early identification of unhealthy aging, interventions for the timely regulation of the GM are being designed [14].

However, the precise mechanism through which the GM changes with age remains unknown, obfuscating changes in the bacterial communities due to the great variety of factors that impact their modification. Nevertheless, these changes are clearly associated with the development of chronic inflammation, neurodegeneration, cognitive decay, and frailty, leading to the use of microbial profiles as biomarkers for geriatric care [8,14].

### 1.3. Inflammation

Aging corresponds to the appearance of low-level chronic inflammation called “inflammatory aging”, i.e., cellular senescence. On the one hand, inflammatory aging is linked to a great variety of diseases associated with age, such as obesity, diabetes type 2, and cardiovascular disease [18]. Meanwhile, cellular senescence is the state of the response given to stress in the cell with accompanying morphological and biochemical changes [19,20,21]. 

Importantly, cellular senescence influences telomere attrition, the mechanisms related to stress and reactive oxygen species, radiation, agents that damage DNA, and the responses of unfolded proteins [21]. The observed effects derived from cellular senescence include growth arrest and replication, resistance to apoptosis, chromatin remodeling, and metabolic reprogramming, among others [22].

Senescent cells also secrete a set of cytokines and chemokines, among other inflammatory mediators such as growth factors, proteases, lipids, and insoluble proteins, as well as components of the extracellular matrix called the Senescence-Associated Secretory Phenotype (SASP), which constantly fosters chronic inflammation [21,22,23]. As a result, cellular senescence through the SASP contributes to the rise of multiple chronic health issues associated with inflammation such as atherosclerosis, osteoarthritis, cancer, Alzheimer’s disease, and insulin resistance [22].

Recently, the relationship between senescence and the diversity of the GM has been considered because changes in the diversity of the microbiota alongside alterations in the function of the intestinal barrier are critical factors in the growth of inflammation, resulting in the aging process [23,24]. Consequently, the elevated presence of inflammatory mediators decreases the appearance of tightly bound proteins, enhancing intestinal permeability and potentially maintaining an inflammatory state [5].

Gut dysbiosis affects the function and synthesis of metabolites, terminating the leakage of proinflammatory microbial products into the bloodstream. This process also positively regulates molecules such as TNF-α, IL-6, and IL-1, which encourage a state of chronic inflammation and SASP [23]. Some authors noted that dysbiosis leads to a vicious proinflammatory cycle that produces or worsens the progression of pathological issues [18]. Indeed, the father of immunity and Nobel prize winner Elie Metchnikoff claimed that senescence arises from metabolites produced by gut bacteria, emphasizing the relevance of the GM in the aging process [25]. However, the relationship within humans remains unclear when considering changes in the GM and the low degree of inflammation in animal models with microbial profiles that are modified in the presence of inflammation. These results suggest that microorganisms have exceptional adaptations to inflammatory states, independent of one’s health [5]. 

Therefore, dysbiosis in the GM among older adults might boost inflammatory aging. Although there are some limitations in this field of study, the liberation of inflammatory cells may be related to intestinal dysbiosis and serve as a catalyst for neuroinflammation, causing cognitive decay or even facilitating the development of characteristic conditions of aging such as frailty, sarcopenia, reduced functional capacity, and mood disorders [19,25].

### 1.4. Neuro-Cognitive Function

It is well known that neurodegenerative diseases involve the progressive loss of neuronal activity, decreasing cognitive function. Reasoning abilities are instrumental in the interdependence of older adults. However, memory, learning, and problem-solving decline with aging [26,27]. 

In addition, impaired cognitive function may play a role in aging due to it leading to complications in the adoption of preventive and therapeutic interventions of specific lifestyle behaviors that have been associated with outcomes such as hospitalization and mortality. Reasons are given as to why impaired neuro-cognitive function has been proposed as a condition that promotes frailty [28].

Despite the multifactorial and complex development pathways involved in this condition, all pathways include the presence of oxidative stress, inflammation, mitochondrial dysfunction, and intracellular Ca^2+^ overload. The influence of the gastrointestinal tract over brain function and vice versa has also been recognized [26,27]. 

The connection between the intestine and the brain has been broadly described and involves not only neurons but also neurotransmitters, hormones, and immunological mediators [29].

Lately, the GM has been implicated as a participant in regulating the gut–brain axis due to communication between the GM and the brain through the vagus nerve and interactions between the immunological system, microbial metabolites, and the endocrine system. Moreover, the GM communicates with the brain by producing neuroactive substances, modulating the intestinal immune system and neurons of the enteric nervous system. Furthermore, the GM fosters the synthesis of neurotransmitters, neuropeptides, and hormones that participate in the brain’s functions and behaviors. This relationship has led to the concept of the microbiota–gut–brain axis, based on the significant interplay between the constituent elements [26,30,31].

Increasing evidence in GM studies and the expansion of neurodegenerative diseases (END) indicate that dysbiosis derives from the modulation of functions and microglia activation since dysbiosis and neuroinflammation are characteristics of END pathology [31]. Notably, microglia represent almost 10% of the most innate immune cells in the central nervous system, contributing to the regulation of numerous biological functions, in addition to actively and quickly identifying and responding to environmental perturbations for homeostasis restoration [32]. 

These factors suggest that dysbiosis generates a loss in the homeostasis of the gut–brain axis, thereby contributing to the physiopathology of END [30]. Studies using animal models have shown the relationship between gut bacterial communities and positive behaviors. A study performed on rats demonstrated the normalization of aggressiveness with the presence of *Lactobacillus fermentum* after the rats received ampicillin as a medication for one month. Another study found that *Lactobacillus helveticusNS8* improved stress in rats by reducing the plasma levels of corticosterone and adrenocorticotropic hormones. This bacterium also raised the levels of serotonin and norepinephrine. In addition, *Bifidobacterium Longum* improved the reasoning functions of mice [33].

Aging itself is a risk factor for the advancement of END because of the physiological changes that occur during this process (considering the composition of the GM). Moreover, bacteriophages have been reported to be regulators of aging and neurodegeneration. These bacteriophages have the potential to vary the balance of the GM by inducing changes in the abundance of specific bacteria. Additionally, bacteriophages can maintain the integrity of the intestinal barrier while adding to the mucous layer, thus controlling invasive bacteria [27,33].

Dysbiosis in the GM increases intestinal permeability with aging, which relates to inflammation that encourages the synthesis and discharge of bacterial components, e.g., LPS, lipoproteins, and bacterial RNA. This process contributes to activation of the immunological system and the release of proinflammatory cytokines (TNF-α, IL-1β, and IL-6) [27,30]. 

Conversely, gut bacteria produce neurotransmitters such as GABA, acetylcholine, and dopamine. In addition, neurotrophic factors such as the neurotrophic factor derived from the brain (BDNF) and the resulting nerve growth factor (NGF) are essential in the transmission of nervous energy. *Lactobacilli* and *Bifidobacteria* were found to help relieve symptoms similar to anxiety and depression when converting glutamate from the intestine into GABA. It was also demonstrated that the quantity of Alcaliganeceae and Porphyromonadaceae is positively correlated with a decrease in reasoning [27,33]. 

Nonetheless, research in the field of the microbiota–gut–brain axis remains unresolved since the omnipresence of the GM and its influence in the physiological system problematizes individual perspectives related to the unique microbiota–gut–brain axis. To date, most studies carried out in this area have used limited animal models, with many relying upon observations in clinical environments, highlighting the need for multidisciplinary and multi-systematic investigations to describe relevant mechanisms and opportunities [29,31]. Consequently, this research will aid in applying biomarkers, such as the IM, to the early detection of neurogenerative processes [34].

### 1.5. Sarcopenia

As previously stated, frailty is a physical condition related to age and is characterized by homeostatic alterations in the organism. Consequently, we must consider all physical manifestations of frailty in which muscular health plays an essential role [35].

Sarcopenia refers to the loss of mass, strength, and function of skeletal muscle derived from the aging process. Factors such as lifestyle, nutrition, hormonal changes, and physical activity influence this condition and increase the risk of falling, fractures, and mortality [34].

While the underlying mechanisms remain the foci of studies on sarcopenia, investigations suggest that the relationship between this process and the GM represents a risk factor for the development of sarcopenia, especially when considering the clear abundance of microbial strains. The GM can affect the quantity and function of muscular mass when it intervenes in the processes of inflammatory regulation, systemic immunity, the metabolism of substances and energy, and sensitivity to insulin. Furthermore, the GM can influence the synthesis of muscular proteins through dependent metabolites, which represent possible substrates for the appearance of sarcopenia [24,35,36].

Aging not only leads to observable changes in the muscles but also modifies bacterial diversity, privileging SCFA species with decreased availability of beneficial bacterial metabolites [37,38].

The GM participates in the metabolism, absorption, and viability of amino acids due to proteins in the diet being hydrolyzed by enzymes for both the host and the bacteria. Although there is evidence that dysbiosis could increase the necessity of proteins for muscular synthesis, a process termed “anabolic resistance”, dysbiosis is also responsible for reducing the synthesis of muscular proteins and the constant alteration of muscular physiology that results in the development of sarcopenia [35,37]. Therefore, the modifications in the GM caused by aging can influence reductions in the production of SCFA, similar to the results when modulating the production of ATP and influencing the deposition of skeletal muscle protein through the regulation of systemic anabolic/catabolic balance. The concept of the “intestine–muscle axis” was established to estimate the glucose uptake in skeletal muscle and the effects of insulin sensitivity and inflammation [35,38].

Diverse studies have connected the decrease in species such as *Firmicutes*, *Akkermansia*, *Ruminococcus*, and *F. prausnitzi* in older adults to the development of sarcopenia and frailty. This phenomenon can be explained by the mechanisms that affect the synthesis of muscular proteins impacting muscular mass [24,32]. Some authors have observed a reduction in *Lactobacilli*, *Bacteroides*/*Prevotella*, and *Faecalibacterium prausnitzii* and an increase in *Ruminococcus*, *Atopobium*, and *Anterobavvteriacae* among senior citizens with high frailty scores. Additionally, a higher abundance of butyrate-producing species such as *Faecalibacterium prausnitzii* was observed in those with adequate functionality, corroborating the impact of bacteria on muscular function [39]. Therefore, while the relationship between the intestines and muscles remains unclear, increasing evidence is laying the foundation for future research on the intestine–muscle axis [40].

The above evidence highlights the importance of consuming pre- and probiotic supplements to compensate for some changes in the GM associated with age [39]. A notable improvement in muscular mass, strength, and resistance capacity was observed after supplementation with *Lactobacillus* and *Bifidobacterium* in both mice with advanced age and senior citizens who participated in clinical studies [38]. Other studies on an elderly patient 65 years of age showed an increase in grip strength after 13 weeks of supplementation with prebiotics. Meanwhile, other studies demonstrated that supplementation with *Faecalibacterium prausnitzi* could increase muscular mass through improvements in mitochondrial respiration [36]. Furthermore, supplementation with *Lactobacillus reuteri* and *Lactobacillus gasseri* restored the balance of the GM, in addition to reducing serum proinflammatory cytokine levels and increasing muscular mass [40]. 

### 1.6. Interventions for Modulation of the Gut Microbiota

As previously noted, the GM is very sensitive to a variety of adverse factors such as lifestyle, diet, sleep deprivation, circadian rhythm disturbances, pharmaceuticals, the social environment, and sedentism [32]. As a result, it is possible to make changes in the quantity and variety of bacteria without improving the specific aspects of one’s lifestyle that facilitate advancements in aging-related comorbidities [15,41].

Diet is one of the most crucial aspects in the composition of the GM because it determines the substrate supply for growth and maintenance, intestinal transit time, and intestinal environmental conditions [30]. Likewise, nutrition influences genetic coding and metabolism for both the GM and the host, creating a symbiotic relationship between the two [41].

On the one hand, the Mediterranean diet is associated with benefits related to a lower risk of developing cardiometabolic diseases and lower frailty. The same diet is correlated with the greater production of SCFA (principal metabolites of the IM) and increasing beneficial bacteria such as *Parabacteroides*, *Bacteroides*, *Christensenellaceae*, and *Methanobrebrevibacter* through the consumption of additional fruits, vegetables, and fiber. On the other hand, diets lower in FODMAPS (oligo-, di-, and monosaccharides and fermentable polyols) and gluten-free diets could affect the GM’s balance, thereby ameliorating cognitive functions. The evidence indicates that consuming dietary fiber regulates the quantity of microorganisms, as well as their metabolites. For instance, the intake of fruits and galacto-oligosaccharides increases the abundance of *Bifidobacterium* and *Lactobacillus* [27]. 

Populations older than 65 years fed with low-fat, high-fiber diets exhibited greater bacterial diversity than participants fed more moderate high-fat and low-fiber diets [38]. In addition, the consumption of polyphenols promotes the growth of a healthy GM. An increase in the diversity of *Bifidobacterium*, *Lactobacillus*, *Akkermansia*, *Christensenellaceae*, and *Verrucomicrobia* has also been associated with possible anti-aging effects. Moreover, the consumption of anthocyanins enhanced the concentrations of *Bacteroidetes* such as AGGCC and decreased *Firmicutes*, generating possible anti-aging effects due to the production of these key metabolites with the potential to change the GM and modify metabolic regulation [42].

The modifications observed in the GM among institutionalized senior citizens focused mainly on decreased levels of healthy bacteria. It is necessary to apply nutritional strategies to prevent such losses. Implementing probiotics and prebiotics is crucial in GM regulation as part of an essential strategy to promote healthy aging.

Interventions with probiotics have yielded improvements in the interactions among essential components of the GM [41]. Probiotics such as *Lactobacillus*, *Bifidobacterium*, *S. thermophilus*, *Enterococcus*, and *Bacillus* facilitated the liberation of neurotransmitters when increasing the levels of tryptophan-derived neurotoxic factors, thereby promoting their administration in the prevention and early treatment of diseases related to cognitive function [27]. 

Nonetheless, studies on supplementation with probiotics have provided contradictory results when controlled for prescription conditions such as dosage, strain, and duration of supplementation, as well as patient factors such as age, current diseases, use of drugs (mainly antibiotics), diet, and lifestyle [41].

Prebiotics are used as fuel for gut bacteria and facilitate the production of AGGCC since they enhance the growth of bacteria such as *Bifidobacterium* and *Lactobacillus* [41]. Prebiotics have yielded improvements in aging, highlighting the importance of insulin and fructooligosaccharides. In a previous study, supplementation using prebiotics produced increased strength and muscular resistance [39].

Polyunsaturated fatty acids such as omega 3, docosahexaenoic acid (DHA), eicosapentaenoic acid, and docosapentaenoic acid are found in foods such as fish, eggs, seafood, and vegetable oils. Fatty acids were linked to an increase in bacterial diversity among senior women, with an improvement in the effects when probiotic strains were simultaneously consumed [41].

However, some differences have also been observed in the GM composition among those with different degrees of physical activity. This situation highlights the favorable impact of the GM. Comparisons were performed among populations of European senior citizens whose levels of physical activity were high or low. Those with higher levels of physical activity were found to have a healthier GM containing *Faecalibacterium prausnitzii*, which was associated with beneficial results. A similar case was examined with *Bifidobacterium adolescente* and *Christensenella*, which are species linked with high levels of physical activity and significant health benefits [43]. 

A previous study found that aerobic exercise increased the sensitivity and quantity of *Bacteroides*, muscular strength, and cardiopulmonary function in women older than 65 after 12 weeks of monitoring [36]. Moreover, such cardiorespiratory improvements could be gained by walking for only 6 min [43].

Although interventions have been explored to modify the composition of the GM during aging, the role of diet, bioactive supplements, and exercise remains scarce in research. Thus, we emphasize the importance of future research to explore this type of alternative therapy in combination with drugs [42].

## 2. Conclusions

Healthy aging provides a guideline to compensate for the physiological and social consequences of this process, thereby improving senior citizens’ quality of life. Current research indicates that the quantity and types of bacteria in the gut microbiota could be considered biomarkers of longevity; such markers could be used to promote healthy aging. Bacteria in the GM help maintain the balance of inflammatory and anti-inflammatory functions as part of the microbiota–gut–brain axis through genetic and epigenetic regulation and various nutritional processes. In addition, bacteria are involved in distinct aspects associated with aging, such as inflammation, immunity, muscular state, and frailty. The identified strains and metabolites associated with aging could be used to design treatments aimed at modifying the GM to improve senior citizens’ quality of life. Based on our results, this field represents an opportunity to develop interventions to enhance healthy aging.

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
