# Peer review of "Impact of Gut Microbiota on Aging and Frailty: A Narrative Review of the Literature"

_geriatrics, 2024, doi:10.3390/geriatrics9050110_

Round 1

Reviewer 1 Report

Comments and Suggestions for Authors

The authors reviewed the recent advances in gut microbiota science in the field of aging. The present study was well organized and well investigated, and will give us a new information especially in the field of aging. To improve the quality of this paper, the authors should revise it according to the following suggestions;

1) In this paper, the authors use the term intestinal microbiota (IM) instead of gut microbiota, but I don't really understand the difference. The title of the paper is gut microbiota.

2)In aging research, there is a debate about how to measure biological age. A recent paper claims that information on gut microbiota is useful and can help predict frailty (Wang H, et al. Gut Microbes 2024, 16: 2297852). A session on this topic should be added.

3) The word fragility used in this paper should also be explained to explain the difference from frailty. Frailty is generally easier to understand.

Author Response

Comments 1: In this paper, the authors use the term intestinal microbiota (IM) instead of gut microbiota, but I don't really understand the difference. The title of the paper is gut microbiota

Response 1:The term intestinal microbiota (IM) was changed with gut microbiota (GM) according to the title of the paper

Comments 2: In aging research, there is a debate about how to measure biological age. A recent paper claims that information on gut microbiota is useful and can help predict frailty (Wang H, et al. Gut Microbes 2024, 16: 2297852). A session on this topic should be added.

Response 2: The reference suggested (Wang H, et al. Gut Microbes 2024, 16: 2297852), was included to justify de relevance of the association between gut microbiota and frailty.

Comments 3: 3) The word fragility used in this paper should also be explained to explain the difference from frailty. Frailty is generally easier to understand. 

Response 3: There was a translate mistake, reason why the term “fragility” was change to “frailty”, which is the right term according to the definition we give on the paper: “frailty is a clinical condition considered to be the highest clinical state of vulnerability of the individual to endogenous and exogenous stressors. It is associated with an increased risk of disability, which is provoked by the predisposition to failure in homeostasis after a stressful event.”

Reviewer 2 Report

Comments and Suggestions for Authors

This review was about the gut microbiota on aging and fragility. The manuscript should be major revision before publication. The specific revision was as follow.

1. The content of 2. Materials and Methods should be deleted.

2. The title 3. Results should be deleted.

3. The content of the manuscript should be consistent with the title. The reason of fragility should be reviewed and the key factor, which the relative intestinal microbiota diversity.

4. The logical order of articles should be rearranged.

5. In the section of “3.2. Gut microbiota in aging” the content As has been described before, the old adults reveal a higher prevalence of Bacteroidetes in comparison with the abundance of Firmicutes observed in younger adults.” means that the Bacteroidetes could increase the prevalence. However , in the section of 3.6 “Moreover, the consumption of anthocyanins enhanced the concentrations of Bacteroidetes like AGGCC and diminished Firmicutes.  which mean the beneficial role of Bacteroidetes. Can you explain the reason?

6. What is the relationship of the “Neuro -cognitive Function” and the fragility, even the intestinal microbiota diversity?

Comments on the Quality of English Language

None

Author Response

Comments 1: The content of 2. Materials and Methods should be deleted.

Response 1: The content of “2. Materials and Methods” was deleted

Comments 2: The title “3. Results ”should be deleted.

Response 2: The title “3. Results” was deleted

Comments 3:  The content of the manuscript should be consistent with the title. The reason of fragility should be reviewed and the key factor, which the relative intestinal microbiota diversity

Response 3: The term “fragility” was changed to “frailty” to unify the content of the manuscript, according to the title

Comments 4: The logical order of articles should be rearranged.

Response 4: The manuscript was ordered

Comments 5: In the section of “3.2. Gut microbiota in aging” the content “As has been described before, the old adults reveal a higher prevalence of Bacteroidetes in comparison with the abundance of Firmicutes observed in younger adults.” means that the Bacteroidetes could increase the prevalence. However , in the section of 3.6 “Moreover, the consumption of anthocyanins enhanced the concentrations of Bacteroidetes like AGGCC and diminished Firmicutes.”  which mean the beneficial role of Bacteroidetes. Can you explain the reason?

Response: The section “Moreover, the consumption of anthocyanins enhanced the concentrations of Bacteroidetes like AGGCC and diminished Firmicutes.”, was complemented with the justify in accordance with the reference used

Comments 6: What is the relationship of the “Neuro -cognitive Function” and the fragility, even the intestinal microbiota diversity?

Response 6: It is developed the explanation of the relationship between neuro-cognitive function and frailty

Round 2

Reviewer 2 Report

Comments and Suggestions for Authors

None

Author Response

The authors thank the reviewer for his/her suggestions.